# Serotonin Levels in the Serum of Persons with Onchocerciasis-Associated Epilepsy: A Case-Control Study

**DOI:** 10.3390/pathogens10060720

**Published:** 2021-06-08

**Authors:** Melissa Krizia Vieri, An Hotterbeekx, Michel Mandro, Joseph Nelson Siewe Fodjo, Alfred Dusabimana, Francoise Nyisi, Deby Mukendi, Joe Gwatsvaira, Samir Kumar-Singh, Robert Colebunders

**Affiliations:** 1Global Health Institute, Faculty of Medicine and Health Sciences, University of Antwerp, 2610 Antwerp, Belgium; Alfred.dusabimana@uantwerpen.be (A.D.); robert.colebunders@uantwerpen.be (R.C.); 2Global Health Institute, Gouverneur Kinsbergencentrum, University of Antwerp, Doornstraat 331, 2610 Wilrijk, Belgium; 3Molecular Pathology Group, Laboratory of Cell Biology & Histology, Faculty of Medicine and Health Sciences, University of Antwerp, 2610 Antwerp, Belgium; samir.kumarsingh@uantwerpen.be; 4Provincial Health Division Ituri, Ministry of Health, P.O. Box 57 Bunia, Democratic Republic of the Congo; michel.mandro@student.uantwerpen.be; 5Brain Research Africa Initiative (BRAIN), Yaoundé P.O. Box 25625, Cameroon; JosephNeson.SieweFodjo@uantwerpen.be; 6Centre de Recherche en Maladies Tropicales, Rethy, P.O. Box 143 Bunia, Democratic Republic of the Congo; fnyisi@gmail.com; 7Centre Neuro-Psycho Pathologique, University of Kinshasa, P.O. Box 127 Kinshasa, Democratic Republic of the Congo; debymukendi@yahoo.fr; 8Department of Mathematics and Statistics, University of Limerick, V94 T9PX Limerick, Ireland; joweygwatsv@gmail.com

**Keywords:** epilepsy, nodding syndrome, onchocerciasis, *Onchocerca volvulus*, serotonin, pathogenesis

## Abstract

Onchocerciasis-associated epilepsy (OAE) is a devastating childhood disorder occurring in areas with high *Onchocerca volvulus* transmission. Despite epidemiological evidence showing the association between *O. volvulus* and epilepsy, the underlying mechanism remains unknown. Since high levels of serotonin are known to induce seizures, we investigated serotonin levels in persons with OAE and controls selected from the Democratic Republic of Congo. Serum serotonin levels were determined by ELISA in 19 persons with OAE, 32 persons with epilepsy without *O. volvulus* infection, 18 with *O. volvulus* infection but without epilepsy, and 35 with neither *O. volvulus* infection nor epilepsy. *O. volvulus* infection was diagnosed by skin snip testing and/or OV16 antibody detection. Serum serotonin levels were significantly decreased in persons with OAE compared to persons with *O. volvulus* infection and no epilepsy. In conclusion, an increased serotonin level is unable to explain the pathogenesis of OAE. Other hypotheses to identify the causal mechanism of OAE will need to be investigated.

## 1. Introduction

A high prevalence of epilepsy, including nodding syndrome, has been reported in onchocerciasis-endemic areas with past or ongoing *Onchocerca volvulus* transmission. This form of epilepsy is characterized by the onset of seizures without any obvious cause in previously healthy children between the ages of 3–18 years. It is now called onchocerciasis-associated epilepsy (OAE) because of its epidemiological association with onchocerciasis [1,2]. Nodding syndrome is one of the most typical clinical presentations of OAE [1]. However, despite the epidemiological evidence for the association with onchocerciasis, the pathological mechanism of OAE remains to be elucidated [3].

Two cohort studies performed in onchocerciasis-endemic regions in Cameroon showed that the *O. volvulus* microfilarial load predicted the risk of developing OAE [3,4]. The severity of OAE disease was also shown to be dependent on the *O. volvulus* infection load [3,4,5]. These findings suggest that the underlying causative mechanism for OAE is directly linked to the parasitic infection. The hypothesis that *O. volvulus* microfilariae can enter the central nervous system (CNS) by directly crossing the blood–brain barrier (BBB) was recently discarded [6]. *O. volvulus* possesses a wide range of excretory and secretory products that are released into the host environment and interact with the host responses [7]. One potential mechanism triggering OAE could be the crossing of parasite-derived substances across the BBB into the CNS [8].

A biomarker study published in 2017 using non-targeted mass spectrometry on serum from ten *O. volvulus*-infected persons identified 286 known metabolites [9]. The authors identified serotonin amongst the top six highly elevated substances in the serum of *O. volvulus*-infected people from Ecuador and Guatemala, while being completely absent in control serum from the NIH blood bank [9]. Although serotonin is produced at low levels by the human gastrointestinal tract and is stored in platelets [10], it is also an important neurotransmitter in filarial nematodes, including *O. volvulus* [11,12]. Low serotonin levels do not cross the BBB, but high levels are shown to cause BBB breakdown [13]. While low levels of serotonin decrease the threshold for seizures [14], high levels, such as during an overdose of selective serotonin reuptake inhibitors, are known to cause ‘serotonin syndrome’, which is characterized by seizures and muscle wasting, features also observed in OAE [15,16].

To investigate whether serotonin may play a pathogenic role in OAE, we determined serotonin levels in persons with OAE, and three control groups: (1) persons with epilepsy (PWE) without *O. volvulus* infection (no OAE); (2) persons without epilepsy but with *O. volvulus* infection; and (3) persons without epilepsy and without *O. volvulus* infection.

## 2. Materials and Methods

### 2.1. Study Population and Sample Collection

Study participants were selected among participants of larger studies performed in an onchocerciasis meso-endemic region in the Logo health zone in Ituri Province (Democratic Republic of Congo (DRC)) [17,18]; and in Mosango, an onchocerciasis hypo-endemic region in Kwilu Province in the DRC.

In August 2016, a house to house survey in onchocerciasis-endemic villages in the Logo health zone, DRC, documented an epilepsy prevalence of 4.6% [17]. In October 2017, persons with epilepsy (PWE) and their caretakers in this area were asked to participate in a clinical trial to assess the effect of ivermectin on the frequency of seizures in *O. volvulus*-infected PWE [18,19]. A PWE was considered to present OAE if he/she was a previously healthy child, living in an onchocerciasis meso- or hyper- endemic area since at least the age of three 3 years, had developed epilepsy between the age of 3–18 years without any obvious reason [1]. After informed consent was obtained, clinical data were collected using a standardized questionnaire. Serum samples were obtained from blood collected from PWE immediately transferred to the laboratory on the same day for long-term storage at −20 °C. Serum of 19 persons, OV16 seropositive, or with the presence of *O. volvulus* microfilariae in skin snips and meeting the OAE criteria [1] were selected for this serotonin study. In addition, during a rapid epidemiological mapping of onchocerciasis (REMO) in the Logo study area, serum samples were obtained from men without epilepsy older than 20 years with or without *O. volvulus* infection who had lived in the area for more than 10 years.

Ten patients with recent-onset neurological disorders (one with Bell’s palsy, one with HIV infection and crypto meningitis, one with psychosis, two with febrile illnesses, one with unspecified headache, one with tetraparesis, one with bacteremia, one with radiculopathy, and one with an abnormal behavior), six with febrile illnesses, and 12 patients with late-onset of epilepsy admitted to Mosango General Referral Hospital in Kwilu Province, DRC, had been recruited between 2012 and 2015. Patients with neurological disorders participated in a study that was part of the project “Better Diagnosis for Infectious Diseases”. A detailed description of the patient population of this study was previously published [20]. Blood and cerebrospinal fluid samples were collected and stored at −80 °C until analysis. None of the samples of persons with OAE were obtained directly before or after seizures. In Mosango, samples were obtained in persons admitted to the hospital because of seizures, but samples were also not obtained immediately after the seizures.

### 2.2. Diagnosis of O. volvulus Infection

For the diagnosis of *O. volvulus* infection, skin snips were taken from the left and the right iliac crests with a sterile corneo-scleral punch (Holt, 2 mm). Skin snips were placed in a 96-well plate in 0.9% saline solution for 24 h to allow the microfilariae to emerge. A total count was performed by microscopy. An OV16 rapid diagnostic test (OV16 RDT, SD Bioline Onchocerciasis IgG4 rapid test, Abbott Standard Diagnostics, Inc., Yongin, Korea) was performed on all study participants from the Logo health zone. Participants were considered *O. volvulus* positive whether they were positive to the skin snip, to antibodies, or to both. During the REMO study in the Logo health zone, men older than 20 years without epilepsy were examined for the presence and number of palpable nodules, and the OV16 RDT was performed, but no skin snips were taken. Individuals were considered uninfected when no palpable nodules were present and the OV16 RDT was negative, and were considered infected when nodules were present in combination with a positive OV16 RDT. The Mosango participants were tested by OV16 antibody ELISA, as described earlier [21].

### 2.3. Detection of Serotonin

Serotonin was detected using an ELISA (Abnova serotonin ELISA Kit, KA1894, Abnova, Paris, France) according to the manufacturer’s instructions. Samples were acetylated and used undiluted directly in the acylation buffer. All samples and standards were run in duplicate. Serum serotonin concentrations were calculated against a standard curve using 4-parametric logistic regression.

### 2.4. Statistical Analysis

Medians and interquartile ranges (IQRs) were used to describe continuous variables, whereas categorical variables were described using absolute and relative frequencies. Differences between the different groups were calculated using the Kruskal–Wallis test, and if this was significant, post-hoc pairwise comparison was performed using the Bonferroni correction. We used the Wilcoxon rank sum test for paired comparison. To assess the relationship between serotonin level and other factors, multiple linear regression was used using serotonin level as a dependent variable, and age, gender, *O. volvulus* infection, and epilepsy status as predictors. The model assumptions were inspected visually and formally. The Shapiro Wilk test was used to explore the normality of the residuals from a regression model. In the case of non-normality of the residuals, a non-linear transformation was explored to deal with this issue. In the case of a non-linear relation between the continuous variable and serotonin level, a quadratic term was introduced in the model. Data were processed and analyzed using IBM SPSS statistics version 27 (2020), SAS version 9.4 (2020), SAS Institute Inc. Cary, NC, USA, and R version 4.0.2 (2020). A two-sided 5% significance level was used.

### 2.5. Ethical Approval

Ethical approval was obtained from the ethics committee of the University Hospital of Antwerp (24 May 2017, B300201733011), the Institutional Review Board of the Institute of Tropical Medicine of Antwerp, Belgium, and the ethics committee of the School of Public Health of the University of Kinshasa (28 February 2018, ESP/CE/013/2018), DRC.

Written informed consent was obtained from each participant, or from their legal representative for those <18 years and with reduced capacity due to their neurological condition. For minors aged 12–18 years, informed assent was obtained in addition to parental consent.

## 3. Results

### 3.1. Study Population

In total, 53 persons without epilepsy and 51 persons with epilepsy, including 19 persons with OAE, were included in the study (Table 1). The mean age of persons with OAE was 11 years old (standard deviation ± 4.76).

In the Logo health zone, the median serotonin level of PWE without *O. volvulus* infection (265 ng/mL) was significantly higher than the median serum serotonin level of the three other groups (Kruskal–Wallis *p* = 0.002) (Table 1). However, among the Mosango participants, the median serum serotonin level of PWE without *O. volvulus* infection was significantly lower than the median serum serotonin level of persons without epilepsy without *O. volvulus* infection (Wilcoxon rank *p* < 0.001) (Table 1). Considering the entire study population, serotonin levels were not significant different across the groups (Kruskal–Wallis *p* = 0.18) (Figure 1A) and there was no significant difference in serum serotonin concentrations between *O. volvulus*-infected persons and non-infected persons (*p* = 0.45) (Figure 1B).

### 3.2. Serotonin Level Is Inversely Correlated to the Presence of O. volvulus

A multiple linear regression including gender, age, province, *O. volvulus* infection, and epilepsy revealed that the mean serotonin level among females was approximately 6.7 (2.587^2^) units higher than the serotonin level among males. In contrast, the mean serotonin level among the persons with epilepsy was approximately 8.8 (2.967^2^) units lower than the serotonin level of the persons without epilepsy. No statistically significant association was observed between serotonin level and *O. volvulus* infection, age, and province (Table 2).

## 4. Discussion

Based on our study results, we need to reject the hypothesis that serotonin could play a pathogenic role in causing OAE. Persons with OAE had the lowest serotonin level of all of the study groups. Moreover, in a multivariable analysis, epilepsy was associated with a lower level of serotonin without an obvious reason for it. Serotonin may be implicated in the regulation of seizures, but whether or not it can potentiate the effects of epileptogenic factors has not been fully established. Animal studies have shown that both an increase and decrease of brain serotonin may decrease the threshold for seizures [13,14,22].

Females had higher serotonin levels. As far as we know, this has not been reported in the literature. What has been demonstrated is that the mean rate of serotonin synthesis in the brain in normal males was found to be 52% higher than in normal females [23].

The strength of our study is that we determined serotonin levels in 19 persons with OAE and in several control groups. However, our study has also several limitations. First, all study groups were small with significant age differences across the groups. Therefore, observed differences in serum serotonin need to be interpreted with great caution. *O. volvulus* diagnosis was also based on different diagnostic tests in the different study groups, and skin snips were not taken from all individuals. In addition, study groups were from two different study sites and this may have influenced the study results. In Mosango, samples were obtained in persons admitted because of seizures, but when exactly the samples were obtained after the seizures was not recorded. Since serotonin levels can fluctuate, it might be that serotonin levels change directly before/after seizures. Finally, in persons with epilepsy, no imaging or laboratory studies were done to investigate the cause of the epilepsy. Nevertheless, our data do not show any evidence that serotonin could play a causal role in OAE. Moreover we cannot confirm that persons with onchocerciasis have a higher level of serotonin [9]. Our results are not in contradiction with the study of Bennuru et al., who detected serotonin in *O. volvulus*-infected individuals. Indeed, this was a metabolic profiling study that used stored serum of only 10 persons with an *O. volvulus* infection from Ecuador and Guatemala and 10 sera from the NIH blood bank [9]. No clinical data of the *O. volvulus*-infected persons were available. Moreover, serum serotonin levels were not measured using ELISA; only serum levels were compared with other metabolites with a mass spectrometry system.

## 5. Conclusions

In conclusion, our study shows that an increased serotonin level is unable to explain the pathogenesis of OAE. Whether a lower level of serotonin may increase susceptibility for epilepsy needs further investigation. Other hypotheses to identify the causal mechanism of OAE will need to be investigated. It is important to discover the mechanism(s) underlying OAE because as long as it remains unclear why children in onchocerciasis-endemic areas develop epilepsy, public health decision-makers may not accept that *O. volvulus* is able to trigger epilepsy, and therefore appropriate preventive interventions (strengthening onchocerciasis elimination efforts in the affected areas) will not be implemented. Moreover, if the cause of OAE is not clearly explained to the people living in these remote onchocerciasis-endemic areas, they will continue to consider epilepsy to be caused by evil spirits, and affected persons and families will remain stigmatized and discriminated against.

## Figures and Tables

**Figure 1 pathogens-10-00720-f001:**
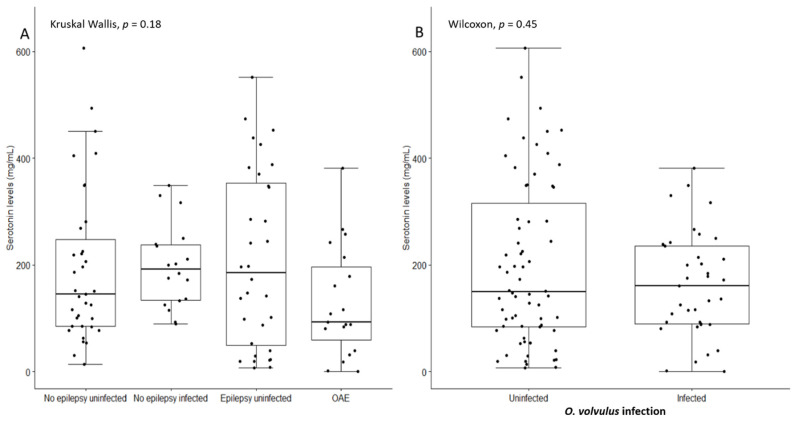
(**A**) Serum serotonin levels in all the persons without epilepsy and with *O. volvulus* infection, persons without epilepsy without *O. volvulus* infection, persons with epilepsy without *O. volvulus* infection, and persons with onchocerciasis-associated epilepsy (OAE). (**B**) Serum serotonin levels in all *O. volvulus*-infected individuals compared to uninfected individuals.

**Table 1 pathogens-10-00720-t001:** Description of the study population and serum serotonin levels.

Logo Health Zone, Ituri Province, Democratic Republic of Congo				
	No Epilepsy ^a^		Epilepsy	
	OV − (*N* = 19)	OV + (*N* = 18)	OV − (*N* = 20)	OAE, all OV + (*N* = 19)
Age in years (median (IQR))	29 (22–37)	47 (35–59)	16 (8–21)	20 (14–28)
Female (*N*, %)	0 (0%)	0 (0%)	13 (75%)	8 (42%)
Nodding seizures (*N*, %)	NA	NA	3 (1.5%)	5 (2.5%)
Serotonin (ng/mL; median (IQR))	116 (62–206)	192 (131–241)	265 (143–386)	92 (39–214)
Mosango, Kwilu Province, Democratic Republic of Congo				
	No Epilepsy ^b^			Epilepsy
	OV − (*N* = 16)			OV − (*N* = 12)
Age in years (median (IQR))	37 (27–48)			26 (19–40)
Female (*N*, %)	8 (50%)			8 (67%)
Serotonin (ng/mL; median (IQR))	203 (106–391)			25 (19–169)

^a^ Healthy volunteers without neurological condition. ^b^ Neurological disorder or febrile illness.

**Table 2 pathogens-10-00720-t002:** Multiple linear regression to assess the factors associated with the square root of serotonin level.

Parameter	Estimate	95% CI		*p*-Value
*O. volvulus* vs. no *O. volvulus* infection	−1.119	−3.494	1.256	0.356
Age (years)	−0.052	−0.127	0.023	0.174
Gender (female vs. male)	2.587	0.207	4.967	0.033
Persons with epilepsy vs. persons without epilepsy	−2.967	−5.413	−0.522	0.017
Place (Ituri vs. Mosango)	1.903	−0.726	4.531	0.156

## Data Availability

The datasets generated during the current study are available from the corresponding authors on reasonable request.

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
