# Peer review of "Serotonin Levels in the Serum of Persons with Onchocerciasis-Associated Epilepsy: A Case-Control Study"

_pathogens, 2021, doi:10.3390/pathogens10060720_

Round 1

Reviewer 1 Report

I thank the authors to clarify some issues that I raised during the first review. However, in my opinion the conclusion remain questionable.

Major comments:

1) The authors state that "despite its potential link with O. volvulus infection, serotonin is unlikely to be involved in the pathogenesis of OAE", but in section 3.2 it is also shown that lower serotonin levels are correlated to epilepsy per se and also gender might play a role. Moreover, OAE cases have lower serotonin levels compared to no epilepsy O. volvulus-infected individuals (Figure 1A). In addition, Figure 1b do not show a link between serotonin levels and O. volvulus infection.

2) In my opinion the authors need to state when the samples were obtained and if this was comparable for all samples, since serotonin levels can fluctuate. It might be that serotonin levels change directly before/after seizures.

3) Figure 1A: p= is missing but since Kruskal-Wallis test is not significant post-hoc test is not needed unless the authors want to highlight and compare two distinct groups. Since the samples were obtained from two different study sites did the authors consider to do statistic analysis separately for each study site?

Reviewer 2 Report

The authors fufilled a reviewer's questions.

Author Response

Thank you 

Reviewer 3 Report

Dear Authors

Good evening.

Thank you for the revised version of your manuscript.

According to my analysis of the revised version, you addressed all my comments, as well as the ones from my fellow colleagues of the revision team.

Accept my best regards

Reviewer 1

Author Response

Thank you

Best regards

Round 2

Reviewer 1 Report

Again, I thank the authors for the reply and the detailed answers to my questions. I think with addition of the limitations the manuscript have now improve. However, in my opinion the conclusion in the abstract is still questionable and despite the addition of the discussion section the authors did not change the abstract. The authors still state a potential link of serotonin levels to Ov infections which are not shown in figure 1b or is there any literature that justify this statement.

Please change the significance values in the figures and use a . instead of ,

Author Response

This manuscript is a resubmission of an earlier submission. The following is a list of the peer review reports and author responses from that submission.

Round 1

Reviewer 1 Report

Review of the manuscript “Serotonin levels in serum of persons with onchocerciasis-associated epilepsy: a case control study -vMelissa Krizia Vieri, An Hotterbeekx, Michel Mandro, Joseph Nelson Siewe Fodjo, Alfred Dusabimana, Francoise Nyisi, Deby Mukendi, Samir Kumar-Singh and Robert Colebunders

Dear Authors

Concerning your manuscript, I believe it is an interesting issue at a specific and broad level of parasitic agents and associated pathologies in Humans, namely when infected with such important parasites as Onchocerca volvulus, a very frequent agent of disease in tropical areas, but also a threat for traveling people in endemic areas. And, this type of research/results should have more visibility, namely because the associated pathologies and co-morbidities with parasitic helminthic infections is growing and their pathogenesis must be fully known to address the better therapeutic approach, so your research will improve our knowledge on this sense.

The final text is well written, with a good level of English. And besides what will be pointed out in my review, namely that your manuscript has a great potential to be published, the final decision on the publication of your manuscript depends on the Editor final statement.

Regarding my review and comments, they are as follows:

Keywords

Page 1

If possible, you should find a way of inserting also Onchocerca volvulus .

Material and Methods

Page 3, Line 111 – Write “The Mosango participants were…”

Page 3, Lines 125-126 – Please provide the statistical packages versions’ year.

Results

Pages 3 and 4 – Please avoid in the final version splitting Table 1 between two pages, since it is easier for the reader to have a glance and read the table if located in just one page.

Page 4, Figure 1B – The x axis should have its title in italic like this: O. volvulus infection.

Discussion

Page 5 – Line 175 – Write epilepsy.

References

Pages 6 and 7 – The scientific names of parasites are not written in italic in most, if not all, references. And it should, so proceed accordingly.

Best regards and good luck with your amendments.

Reviewer

Reviewer 2 Report

The article is dealing with serotonin level related to OAE. Then the definition of OAE is extremely important.

Considering that OAE is characterzed by the onset of seizures without any obvious cause, starting between the ages of 3–18 years in previously healthy persons residing in the onchocerciasis endemic area, the authors must provide age of onset of epilepsy in O. volvulus positive persons. In addition, the authors must provide the detailed criteria how 20 patients are categorized to OAE case. 

Reviewer 3 Report

Vieri et al. measured serotonin levels of individuals from two different areas and postulate that serotonin levels are not associated with OAE.

In my opinion, this conclusion cannot be postulated in this manner since sample size is very limited, the samples are mixed from different study areas, different Ov diagnostic (Microfilariae counts vs. Ov16 Ab vs nodules) was performed. As published recently by the very same group the samples from the Mosango area comes from hypoendemic region (prevalent for T. solium) and thus controls are missing and as shown in Table 1 groups are not even comparable (neurological disorder but no epilepsy vs no epilepsy).

In addition, by simply doing an serotonin ELISA without taking other variables like age or gender (MF and nodule counts?) or serotonin synthesis and metabolism-related molecules into account the drawn conclusion of the paper is not clear especially since other publications showed different results.